# Lack of TLR4 modifies the miRNAs profile and attenuates inflammatory signaling pathways

Juan R. Ureña-Peralta[1], Raúl Pérez-Moraga[2,3], Francisco García-García[2], Consuelo Guerri[1]*

1 Molecular and cellular pathology of Alcohol Laboratory, Prince Felipe Research Center, Valencia, Spain, 2 Bioinformatics & Biostatistics Unit, Prince Felipe Research Center, Valencia, Spain, 3 Biomedical Imaging Unit FISABIO-CIPF, Prince Felipe Research Center, Valencia, Spain

* guerri@cipf.es

**Data Availability Statement:** All fataq files are available from the NCBI's Gene Expression

## Abstract

TLR4 is a member of the toll-like receptors (TLR) immune family, which are activated by lipopolysaccharide, ethanol or damaged tissue, among others, by triggering proinflammatory cytokines release and inflammation. Lack of TLR4 protects against inflammatory processes and neuroinflammation linked with several neuropathologies. By considering that miRNAs are key post-transcriptional regulators of the proteins involved in distinct cellular processes, including inflammation, this study aimed to assess the impact of the miRNAs profile in mice cortices lacking the TLR4 response. Using mice cerebral cortices and next-generation sequencing (NGS), the findings showed that lack of TLR4 significantly reduced the quantity and diversity of the miRNAs expressed in WT mice cortices. The results also revealed a significant down-regulation of the miR-200 family, while cluster miR-99b/let-7e/miR-125a was up-regulated in TLR4-KO vs. WT. The bioinformatics and functional analyses demonstrated that TLR4-KO presented the systematic depletion of many pathways closely related to the immune system response, such as cytokine and interleukin signaling, MAPK and ion Channels routes, MyD88 pathways, NF-κβ and TLR7/8 pathways. Our results provide new insights into the molecular and biological processes associated with the protective effects of TLR-KO against inflammatory damage and neuroinflammation, and reveal the relevance of the TLR4 receptors response in many neuropathologies.

## Introduction

Toll-like receptors (TLRs) are a family of pattern-recognition receptors (PRRs) that recognize conserved structural motifs in a wide array of pathogens or PAMPs (pathogen-associated molecular patterns), as well as damaged tissue or DAMPs (damage-associated molecular pattern) [1]. These receptors are expressed in immune cells and TLRs activation leads to the downstream stimulation of different signaling pathways that trigger the production of large amounts of inflammatory proteins and cytokines. These pathways include type I interferons (IFNs), antiviral proteins through the activation of interferon regulatory factor (IRF) 3, IRF7, activator protein-1 (AP-1) and nuclear factor-kappa B (NF-κβ) [2]. In the central nervous

Omnibus database accessible through GEO Series accession number GSE120373.

**Funding:** This work has been supported by grants from the Spanish Ministry of Science and Innovation (SAF2015-69187R)/ (URL: http://www.ciencia.gob.es/portal/site/MICINN/?lang_choosen=en). And the Institute Carlos III and FEDER funds (RTA-Network, RD16/0017/0004)/ (URL: https://www.isciii.es/Paginas/Inicio.aspx) granted to C.G. The funders had no role in study design, data collection and analysis, decision to publish, or preparation of the manuscript.

**Competing interests:** The authors have declared that no competing interests exist.

system, most TLRs are expressed in glial cells [3]. Recent evidence demonstrates the participation of the TLRs response in neurodegenerative disorders [4, 5].

TLR-signaling pathways are strictly regulated at many levels to prevent excessive inflammation and to achieve balanced output. Recently, several studies have demonstrated the regulatory role of miRNAs in the control of the immune response and TLRs to the pathogens and regulators involved in TLR-signaling [6, 7].

MicroRNAs (miRNAs, miRs) are short (17–24bp) RNA species expressed across cell types that are active against a high proportion of the transcriptome [8]. The sequence-complementary mechanism of miRNA activity exploits the combinatorial regulation of gene expression by repressing the translation of their complementary target genes [9]. These small RNAs play a crucial role in the regulation of diverse biological processes, such as tissue development and homeostasis [10], cell proliferation and differentiation, apoptosis, and immune system function [11]. However, dysregulated miRNAs contribute to the development of several diseases, such as cancer, and cardiovascular or neuroinflammatory and neurological diseases [12].

Ethanol is a neurotoxic compound and its abuse can cause neural damage. However, the mechanism of its neurotoxicity remains unclear. We have shown that by activating TLR4 receptors in glial cells [13, 14], ethanol triggers the release of cytokines and inflammatory mediators to cause neuroinflammation and brain damage in mice with chronic alcohol consumption [15]. The critical role of the TLR4 response in the neuroinflammatory effects of ethanol has been further supported by demonstrating that TLR4-deficient mice are protected against ethanol-induced neuroinflammation and neural damage [16]. To explore the regulatory action of miRNAs in the effects of ethanol on the brain and the influence of the TLR4 response, we used high-throughput sequencing methods along with a bioinformatics analysis in the cortices of the TLR4-WT and TLR4-KO mice treated with and without ethanol. The results demonstrate that ethanol treatment induces a differential expression of some miRNAs, such as the miR-183 cluster (miR-183C) (miR-96/-182/-183), miR-200a and miR-200b, which were down-regulated, while mirR-125b was up-regulated in the alcohol-treated WT *versus* (*vs*.) untreated mice. Some of these miRNAs modulate the target genes related to the genes associated with the innate immune TLR4 signaling response (Il1r1, Mapk14, Sirt1, Lrp6 and Bdnf). However, changes in the neuroinflammatory target genes associated with alcohol abuse were abolished mostly in the ethanol-treated TLR4-KO mice [17]. These results raise the question as to what the impact of the inflammatory or anti-inflammatory associated miRNAs profile is on mice lacking the TLR4 response.

Using high-throughput sequencing (NGS) data from the WT and TLR4 knockout mice cortices, along with a bioinformatics analysis, the aims of this study were to evaluate whether: 1) a differential or specific expression exists in certain miRNAs that modulate the inflammatory pathways associated with TLR4; 2) compensatory changes occur in certain anti-inflammatory miRNAs that could explain the reduction in neuroinflammation associated with alcohol intake.

## Material & methods

### Animals

Female wild-type (TLR4_WT, TLR4+/+, WT) (Harlan Ibérica S.L., Barcelona) and TLR4 *knockout* (TLR4_KO, TLR4-/-, KO) mice were used, which were kindly provided by Dr. S. Akira (Osaka University, Japan) with C57BL/6J genetic backgrounds. Animals were kept under controlled light/dark (12 h/12 h) conditions at 23°C and 60% humidity. The animal experiments were carried out in accordance with the guidelines set out in the European

Communities Council Directive (86/609/ECC) and Spanish Royal Decree 1201/2005, and were approved by the Ethical Committee of Animal Experimentation of CIPF (Valencia, Spain).

## Alcohol treatment

For the chronic alcohol treatment, 44 (11 animals/group) 7-week-old WT (C57BL/6J) and TLR4-KO female mice were housed (4 animals/cage) and maintained with either water (WT and TLR4-KO control) or water containing 10% (v/v) alcohol. They were placed on a solid diet *ad libitum* for 5 months. During this period, daily food and liquid intake were similar for both the WT and TLR4-KO mice and the alcohol-treated/untreated groups. Body weight gain at the end of the 5-month period was similar in both the WT (C57BL/6J) and TLR4-KO mice treated with or without alcohol, as previously described [15]. The peak blood alcohol levels (BALs) detected in mice after the chronic ethanol treatment were around ≈125 mg/dl (range of 87–140 mg/dl) in the ethanol-treated WT mice, and ≈ 122 mg/dl (range of 98–135 mg/dl) in the ethanol-treated-KO. The use of females instead of males was based on our previous studies showing that females were more vulnerable to the effects of ethanol than males [15].

## Cerebral cortex dissection

Mice were sacrificed by cervical, brains were removed and cerebral cortices were dissected following the mouse brain atlas coordinates instructions [18]. Brain cortices were weighed and immediately snap-frozen in liquid nitrogen. Samples were stored at -80˚C until processed.

## Total RNA isolation

The frozen cortex samples (100–200 mg) were used for the total and small RNA (sRNA) extractions. Briefly, 100–200 mg of tissue were disrupted with 1 ml of QIAzol (Qiagen, Maryland, USA), followed by the phenol chloroform method [19]. Total RNA and sRNA were isolated using the miRNeasy columns from the Qiagen Kit to obtain a separate sample for each RNA type. sRNAs were used for the deep sequencing protocol. Total RNA was employed for RT-qPCR to evaluate miRNAs and genes.

## DNA isolation and genotyping

The genomic DNA from the WT and TLR4-KO mice was isolated using the commercial Maxwell 16 mouse tail DNA purification kit and the Maxwell 16 Instrument (Promega, Barcelona, Spain). Following the manufacturer's instructions, DNA was collected in 300 μl of elution buffer. DNA was amplified with specific primers designed to differentiate WT and TLR4-KO strains. For genotyping purposes, three primers were designed according to previous studies [20]: primer "*b*", which was recognized by both genotypes (WT and TLR4-KO); primer "a", which was specific for the WT mice; primer "c", which was specific for the TLR4-KO mice. PCR was performed using 2 μl of DNA extract with master mix 2x PCR TaqNova-RED (Blirt, Gdańsk, Poland). The thermocycler (Eppendorf) program was 40 cycles: 30 seg 94 ˚C+ 90 seg 67 ˚C and 60 seg 74 ˚C. Amplicons were loaded in 1.5% agarose gel and visualized in BioRad. The employed primers are described in Table 1.

## RNA quantity and quality determinations

The quantities of each total RNA sample were determined using NanoDrop™, and quantity and qualities were measured in an Agilent 2100 bioanalyzer. Total RNA integrity was analyzed by the RNA Nano6000 kit (Agilent Technologies, Santa Clara, CA, USA) and the sRNA kit

**Table 1. The RT and RT-qPCR primers used.**

| PRIMER | SEQUENCE |
|---|---|
| **PRIMER A** | CGTGTAAACCAGCCAGGTTTTGAAGGC |
| PRIMER B | TGTTGCCCTTCAGTCACAGAGACTCTG |
| PRIMER C | TGTTGGGTCGTTTGTTCGGATCCGTCG |
| PPIA_F/PPIA_R | GCGTCTGCTTCGAGCTGTTTGC / ACATGCTTGCCATCCAGC |
| TLR4_F/TLR4_R | TGCCTCTCTTGCATCTGGCTGG/CTGTCAGTACCAAGGTTGAGAGCTGG |
| TRIL_F/TRIL_R | CCAACGGCAACGAGATTG / CGGTTAGATTCCAGGTGTAGG |
| IL1B_F/IL1B_R | CTCATTGTGGCTGTGGAGAA / TCTAATGGGAACGTCACACA |
| IL6_F/IL6_R | AAGCCAGAGTCCTTCAGAGAGA / TCTTGGTCCTTAGCCACTCCT |
| COX2_F/COX2_R | CATTGACCAGAGCAGAGAGATG / GGCTTCCAGTATTGAGGAGAAC |
| IL1R_F/IL1R_R | TGAAGAGCACAGAGGGGACT / CATTGATCCTGGGTCAGCTT |
| TRAF6_F/TRAF6_R | AACGTCCTTTCCAGAAGTGC / GAATGTGCAAGGGATTGGAG |
| RT-MIR-592-5P | GTCGTATCCAGTGCAGGGTCCGAGGTATTCGCACTGGATACGACACATCATCGA |
| PCR-MIR-592-5P | ACACTCCAGCTGGGATTGTGTCAATATGCG |
| RT-MIR-377-3P | GTCGTATCCAGTGCAGGGTCCGAGGTATTCGCACTGGATACGACACAAAAGTTG |
| PCR-MIR-377-3P | ACACTCCAGCTGGGATCACACAAAGGCAAC |
| RT-MIR-382-3P | GTCGTATCCAGTGCAGGGTCCGAGGTATTCGCACTGGATACGACAAAAAGTGTTG |
| PCR-MIR-382-3P | ACACTCCAGCTGGGTCATTCACGGACAAC |
| RT-MIR-27B-5P | GTCGTATCCAGTGCAGGGTCCGAGGTATTCGCACTGGATACGACGTTCACCAATC |
| PCR-MIR-27B-5P | ACACTCCAGCTGGGAGAGCTTAGCTGATTG |
| RT-MIR-96-5P | GTCGTATCCAGTGCAGGGTCCGAGGTATTCGCACTGGATACGACAGCAAAAATG |
| PCR-MIR-96-5P | ACACTCCAGCTGGGTTTGGCACTAGCAC |
| RT-MIR-1982-3P | GTCGTATCCAGTGCAGGGTCCGAGGTATTCGCACTGGATACGACCTGTGGGAGAAC |
| PCR-MIR-1982-3P | ACACTCCAGCTGGGTCTCACCCTATGTTC |
| RT-MIR-5122 | GTCGTATCCAGTGCAGGGTCCGAGGTATTCGCACTGGATACGACCACAGCCCCGG |
| PCR-MIR-5122 | ACACTCCAGCTGGGCCGCGGGACCCGG |
| RT-MIR-182-5P | GTCGTATCCAGTGCAGGGTCCGAGGTATTCGCACTGGATACGACCGGTGTGAGTTC |
| PCR-MIR-182-5P | ACACTCCAGCTGGGTTTGGCAATGGTAGAAC |
| RT-MIR-183-5P | GTCGTATCCAGTGCAGGGTCCGAGGTATTCGCACTGGATACGACAGTGAATTCTAC |
| PCR-MIR-183-5P | ACACTCCAGCTGGGTATGGCACTGGTAG |
| RT-MIR-429-3P | GTCGTATCCAGTGCAGGGTCCGAGGTATTCGCACTGGATACGACACGGCATTACC |
| PCR-MIR-429-3P | ACACTCCAGCTGGGTAATACTGTCTGGTAAT |
| RT-MIR-141-3P | GTCGTATCCAGTGCAGGGTCCGAGGTATTCGCACTGGATACGACCCATCTTTACCAG |
| PCR-MIR-141-3P | ACACTCCAGCTGGGTAACACTGTCTGGTA |
| RT-MIR-9-5P | GTCGTATCCAGTGCAGGGTCCGAGGTATTCGCACTGGATACGACTCATACAGCTAG |
| PCR-MIR-9-5P | ACACTCCAGCTGGGTCTTTGGTTATCTAGC |
| RT-MIR-136-3P | GTCGTATCCAGTGCAGGGTCCGAGGTATTCGCACTGGATACGACAAGACTCATTTG |
| PCR-MIR-136-3P | ACACTCCAGCTGGGATCATCGTCTCAAATG |
| RT-MIR-99B-5P | GTCGTATCCAGTGCAGGGTCCGAGGTATTCGCACTGGATACGACCGCAAGGTCGGT |
| PCR-MIR-99B-5P | ACACTCCAGCTGGGCACCCGTAGAACCG |
| RT-MIR-125A-5P | GTCGTATCCAGTGCAGGGTCCGAGGTATTCGCACTGGATACGACTCACAGGTTAAAG |
| PCR-MIR-125A-5P | ACACTCCAGCTGGGTCCCTGAGACCCTTTAAC |
| RT-MIR-125B-5P | GTCGTATCCAGTGCAGGGTCCGAGGTATTCGCACTGGATACGACTCACAAGTTAGG |
| PCR-MIR-125B-5P | ACACTCCAGCTGGGTCCCTGAGACCCTAAC |
| REVERSE PRIMER | CCAGTGCAGGGTCCGAGGT |
| TLR7_F / TLR7_R | TGTGGACACGGAAGAGACAA / CCCTCAGGGATTTCTGTCAA |
| TLR8_F/ TLR8_R | AACCATCGTCAACTGCATGA / CATTTGGGTGCTGTTGTTTG |
| IL12_F / IL12_R | ACGGCCAGAGAAAAACTGAA / CTACCAAGGCACAGGGTCAT |
| INFA_F / INFA_R | AGTGAGCTGACCCAGCAGAT / CAGGGGCTGTGTTTCTTCTC |

was employed for sRNAs (Agilent Technologies, Santa Clara, CA, USA). The best nine samples for each condition were selected and combined to obtain three pooled samples, which gave 12 pooled sRNA samples. Briefly for each condition, nine animals were used and divided into three sample pools. With this approach, an attempt was made to minimize differences due to individuals and, in turn, to increase differences due to the studied variables. Then the sRNA profiles were measured again with the small-RNA kit (Agilent Technologies, Santa Clara, CA, USA) following the manufacturer's instructions. Total RNA integrity was measured by the RNA Nano6000 kit (Agilent Technologies, Santa Clara, CA, USA).

## Small RNA library preparation

First 100 ng of the sRNA fraction from the pooled cortex samples were used to prepare the sRNA libraries with the Truseq library prep Small RNA Sample Preparation kit (Illumina, San Diego, USA). These samples were employed for sequencing in HiSeq following the Illumina pooling manufacture's guidelines. The cDNA from miRNAs was obtained by the Superscript II Reverse Transcriptase kit (Thermo Fisher Scientific, Carlsbad, CA, USA) and unique indices were introduced during PCR amplification for 15 cycles. The sRNA libraries were visualized and quantified in an Agilent 2100 bioanalyzer. A multiplexed pool was prepared that consisted of equimolar amounts of sRNA-derived libraries. Libraries were sequenced for 50 single read cycles in HiSeq2000 (Illumina).

## Reverse transcription miRNA (miRNA-RT)

First of all, 500 ng of total RNA from cortical brain tissue were used. Samples were treated with *DNase I* (Invitrogen, Foster City, CA, USA) to avoid genomic DNA contamination. The retrotranscription reaction was run with specific RT-primers (1nM) (Integrated DNA Technologies, Inc.) for each analyzed miRNA using the High Capacity cDNA Reverse Transcription kit (Thermo Fisher Scientific, Foster City, CA, USA) following the manufacturer's protocol; the specific primers are detailed in Table 1. The reaction was carried out in an Eppendorf 5341 Master Cycler (Eppendorf AG, Hamburg, Germany) at 25 °C for 10 min, then at 40 °C for 1 h, and finally at 85 °C for 5 min to inactivate the enzyme. Total RNA was also converted into cDNA. Briefly; 2 μg of total RNA from the cortical brain tissue were retrotranscribed with the High Capacity cDNA Reverse Transcription kit (Thermo Fisher Scientific, Foster City, CA, USA) following the manufacturer's protocol.

## Real-time quantitative PCR

RT-qPCR was performed in a Light Cycler® 480 System (Roche, Mannheim, Germany). The reactions contained *Light Cycler 480 SYBR Green I Master (2X)* (Roche Applied Science, Mannheim, Germany), 5 μM of the forward and reverse primers, and 1 μL of cDNA. The amplification efficiency (E) of the primers was calculated from the plot of the *Cq* values against the cDNA input according to the equation E = [10(-1/slope)]. The relative expression ratio of a target/reference gene was calculated by the Pfaffl method [21]. Housekeeping cyclophilin-A (Ppia) was used as an internal control for messengers and RNAU6 for small RNAs. The primer gene sequence is detailed in Table 1.

## Bioinformatics /pipelines analysis

The preprocessing of reads was done with Cutadapt (version 2.0) [22]. After removing adapters, the trimmed sequences were aligned against the reference genome (GRCm38.pp6) by

Bowtie2 (v. 2.3.5.1) [23]. As the genomic alignment of miRNAs is challenging given their size ranges (~21 nucleotides) [24], Bowtie2 was configured to increase sensitivity [25] (Fig 2A).

A read count was performed by custom R scripts, and these scripts used the library RSubread [26]. In order to annotate the reads aligned against miRNAs, the genome coordinates from the miRBase/gff3 file [27] were used, which allowed the mature miRNA of interest to be detected and annotated. After generating the count matrix of the six samples, the gene expression data were evaluated by multidimensional scaling and clustering methods to detect any abnormal patterns in the samples. Finally, the Top 10 most abundant miRNAs by read counts in the samples were annotated, along with their corresponding GO terms (Biological Process) from the QuickGO database [28] for descriptive purposes.

The count matrix was normalized by the TMM method (Trimmed Mean of M values). The differential expressions were analyzed by the Bioconductor package of edgeR [29]. P-values were corrected from the False Discovery Rate (FDR) as proposed by Benjamini and Hochberg [30].

## Gene Set Enrichment Analysis (GSEA)

The bioinformatics functional analysis was performed using the mdgsa (Multi-Dimensional Gene Set Analysis) package [31], while Cluster-Profiler [32] was utilized for the graphics/results representation. This functional profiling included several steps:

First, miRNA was linked with its targets using TargetScan Mouse (Release 7.2: August 2018). Two types of miRNA-to-Gene lists were used: one inferred by bioinformatic methods and the other validated by the experimental assays.

Second, the differential expression results were used in the mdgsa package to transform the miRNA expression level into a gene level, which allowed the gene inferred differential inhibition score or index to be obtained. This transferred index contained the effect of multiple miRNAs against the same gene. By this approach, it was possible to obtain a miRNA regulation model and the effects of many miRNAs against their target.

Third, the inferred gene index was used in a univariate gene set analysis [33].

The above methodology allowed us to correlate a large set of genes with different functional annotations (GO terms, KEGG and Reactome pathways, etc.). In fact as we herein obtained a ranking of genes with differential inhibition indices, it was possible to determine if a functional annotation was inhibited in either the WT or the TLR4-KO group. The employed functional annotations were Gene Ontology (Biological Process) terms [34], the Kyoto Encyclopedia of Genes and Genomes (KEGG) [35] and the Reactome pathways database [36]. Multiple testing corrections were made with the FDR developed by Benjamini and Hochberg. Data representation was carried out with ClusterProfiler, a Bioconductor package.

## Statistical methods

The SPSS, version 17.0, and the R version 3.4.3 software 40 were used for the validation analysis and bioinformatics, respectively. The RT-qPCR data were analyzed by a Student's t-test when comparing TLR4_WT and TLR4_KO. A two-way ANOVA was used in the ethanol treatment experiments when comparing more than two groups. Differences at a value of $P < 0.05$ were considered statistically significant.

## Results

### Lack of TLR4 attenuates inflammatory pathways

To assess whether lack of the TLR4 receptor would lower the expression of the inflammatory pathways, we firstly checked the genotypes of the WT and TLR4-KO mice. The agarose gel

(see Fig 1A) illustrated that while the WT mice only showed a band that was derived by the amplification with primers "*a*" and "*b*", the TLR4-KO mice only presented amplification when primers "*c*" and "*b*" were utilized, which confirmed the two herein used genotypes. We also proved the absence of mRNA TLR4 in the TLR4-KO mice (Fig 1B).

We next measured the expression of the different genes and proteins involved in the TLR4 pathway. Fig 1B shows that while the gene expressions of Tril (TLR4 interactor with leucine-rich repeats) and interleukin Il-6 lowered in the TLR4-KO cortices, no change or up-regulation was observed in the gene expressions of Traf6 (tumor necrosis factor receptor-associated factor 6) and receptor Il-1r1, respectively, compared to the WT mice.

To further confirm our previous results, which demonstrated that lack of TLR4 attenuated the inflammatory response in TLR4-KO mice, we also evaluated the levels of some inflammatory-related genes in the mice cortices of the WT and TLR4-KO control with ethanol. Fig 1C shows that while ethanol treatment increased the gene expression levels of IL-1R, IL-1β, Il-6 and COX-2 in the WT mice cortices, the same ethanol treatment did not affect the levels of these genes in the TLR4-KO mice with the same treatment. These findings indicate that the inflammatory signaling response did not occur in the TLR4-KO mice cortices.

## Analysis of the mature miRNA's sequences from the WT and TLR4-KO mice cortices

With the deep sequencing analysis data (see Ureña et al., 2018, **GEO Series accession number GSE120373**), we next evaluated the mature miRNAs profile in the WT and TLR4-KO mice cortices. For this purpose, we followed the methodology scheme shown in Fig 2A, where sequences were mapped against the mouse reference genome (GRCm38.pp6) using Bowtie2 aligner (version 2.3.5.1). We also employed the miRNA annotation GFF file from miRbase (Release 22.1), which extracts the reads mapped to mature miRNAs sequences. The read count with R custom scripts and the Rsubread library were used to obtain a yield of 1.264.521 miRNA reads for the six samples.

Fig 2B shows the most abundant mature miRNA in our NGS data, where 50% of the total miRNA found belonged to only six miRNA: mmu-miR-181a-5p, mmu-miR-127-3p, mmu-miR-26a-5p, mmu-miR-434-3p, mmu-miR-22-3p and mmu-miR-128-3p. When focusing on samples separately, the MDS plot identified that both the WT and TLR4-KO samples clustered correctly. It is noteworthy that sample KO1 exhibited an abnormal pattern compared to samples KO2 and KO3. Therefore, we decided to eliminate this outliner sample to perform further bioinformatics analyses. In relation to this action, each individual sample was formed by a pool of three independent cortices.

Table 2 shows the functional annotation for the top miRNAs found. The more representative miRNA in the samples have GO terms that relate to the regulation of the biosynthesis of interleukins (mmu-miR-26a, mmu-miR-128-3p, mmu-miR-181a-5p) or neurodevelopment regulation (mmu-miR-22-3p, mmu-miR-26a, mmu-miR-125a-5p). We also found the positive regulation of microglial cell activation and neuroinflammatory response (mmu-miR-128-3p) or central nervous system myelin maintenance (mmu-miR-26a-5p), among others. These results indicate the major regulation of these routes in mice cortices. All the annotated miRNA has a GO term related to gene silencing by miRNA.

## Lack of TLR4 significantly reduces the quantity and diversity of the miRNAs expressed in the WT mice cortices

Using the information provided by the NGS technology, we analyzed the miRNAs that were either specific for each genotype or commonly expressed in the WT and TLR4-KO mice.

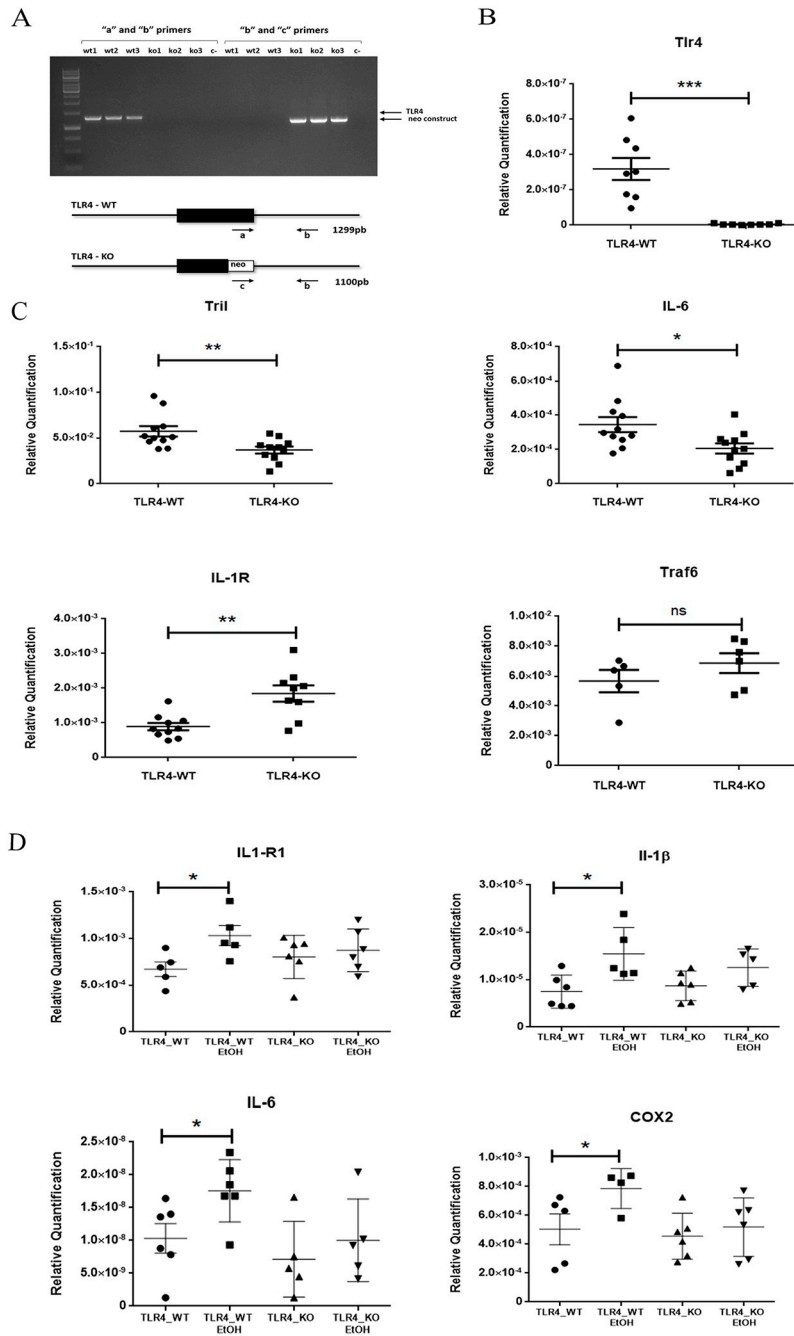

**Fig 1. Lack of TLR4 alters the ethanol-induced immune response in mice cortices.** (**A**) The genotyping analysis of the wild-type (WT) and TLR4KO mice with specific primers for the locus of the TLR4 receptor were confirmed by agarose gel. (**B**) The RT-qPCR analysis in the WT and TLR4-KO mice cortices showed lack of TLR4 gene expression in the TLR4-KO mice. (**C**) The RT-qPCR analysis indicated the deregulation of some TLR4 pathway components, such as Il-6, Tril and IL1R, but not in others like the Traf6 gene. (**D**) The RT-qPCR analysis of IL1R1, IL1β IL6 and COX2 confirmed that ethanol treatment triggered the activation of specific TLR4 signaling pathway components in the WT mice cortices, but ethanol treatment effects were blocked or attenuated in the TLR4-KO mice. A statistical analysis was performed by the t-test for 1B and-1C and a two-way ANOVA for 1D. Asterisk indicates the significance with a p-value $^*<0.05$, $^{**}<0.01$ and $^{**}<0.001$. n = 5–10.

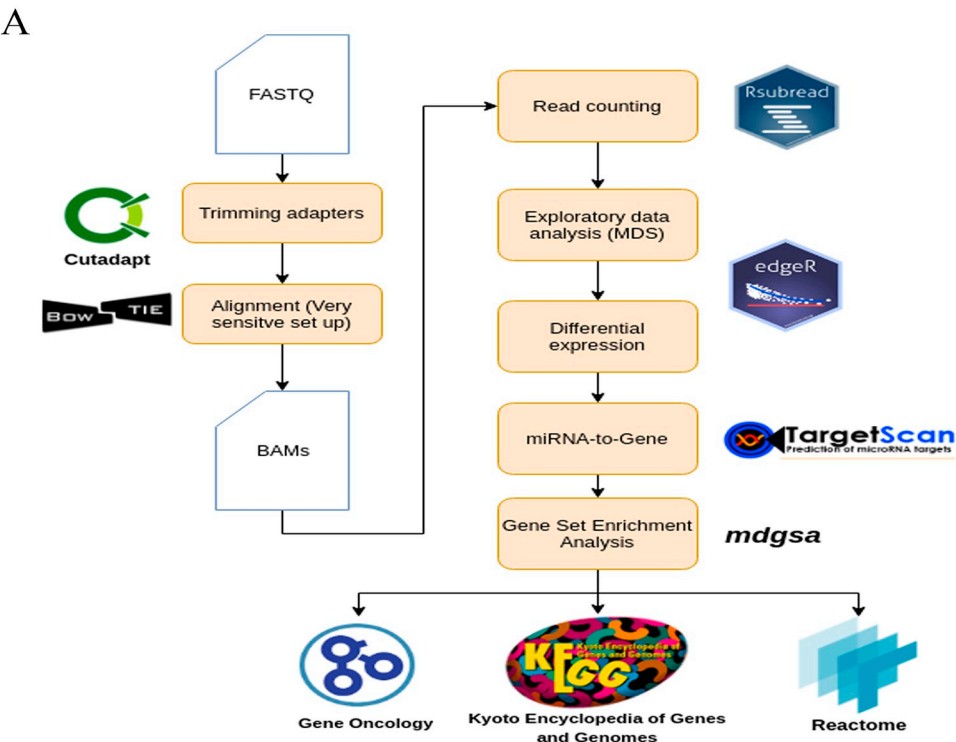

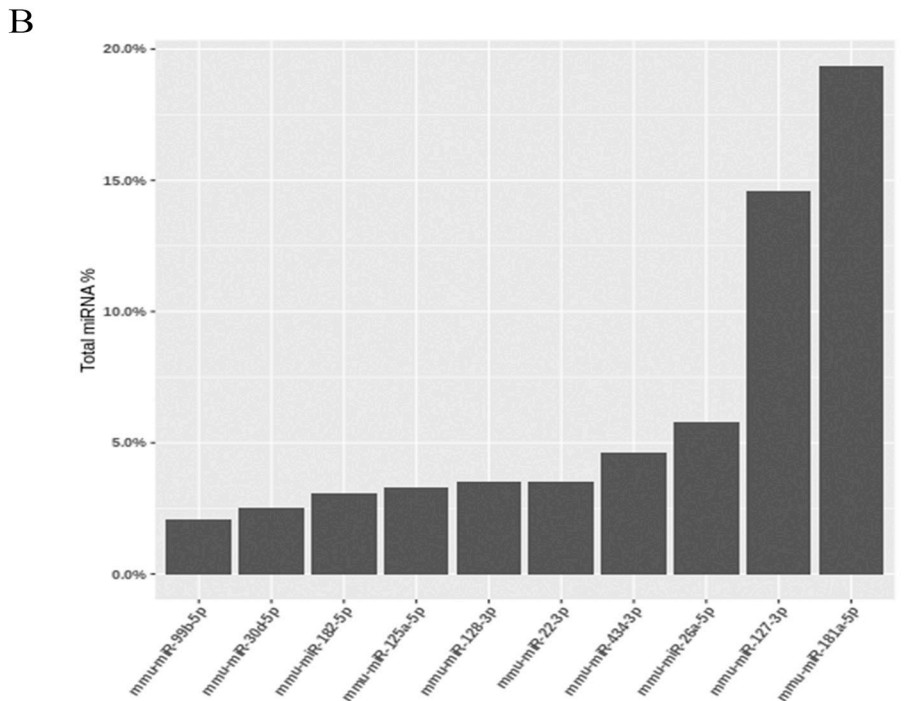

**Fig 2.** Bioinformatics analysis and mature miRNA profile of samples: (**A**) Pipeline methodology used (see page 8). (**B**) The Top 10 abundant mature miRNA in samples, which were almost 50% of the total counts of the following mature miRNA: mmu-miR-181a-5p, mmu-miR-127-3p, mmu-miR-26a-5p, mmu-miR-434-3p, mmu-miR-22-3p and mmu-miR-128-3p.

**Table 2. The most represented mature miRNAs in mice cortices.** The mature miRNAs were annotated with Gene Ontology in relation to a biological process using the QuickGO database [27].

| MIRNA NAME | GO ID | GO DESCRIPTION |
|---|---|---|
| MMU-MIR-22-3P | GO:0035195 | gene silencing by miRNA |
| | GO:0051152 | positive regulation of smooth muscle cell differentiation |
| | GO:2001200 | positive regulation of dendritic cell differentiation |
| MMU-MIR-26A-5P | GO:0010459 | negative regulation of heart rate |
| | GO:0032286 | central nervous system myelin maintenance |
| | GO:0035195 | gene silencing by miRNA |
| | GO:0045379 | negative regulation of the interleukin-17 biosynthetic process |
| | GO:0045409 | negative regulation of the interleukin-6 biosynthetic process |
| | GO:0060371 | regulation of atrial cardiac muscle cell membrane depolarization |
| | GO:0090370 | negative regulation of cholesterol efflux |
| | GO:0150079 | negative regulation of the neuroinflammatory response |
| | GO:1903609 | negative regulation of inward rectifier potassium channel activity |
| MMU-MIR-128-3P | GO:0010985 | negative regulation of lipoprotein particle clearance |
| | GO:0035195 | gene silencing by miRNA |
| | GO:0042632 | cholesterol homeostasis |
| | GO:0055088 | lipid homeostasis |
| | GO:0090370 | negative regulation of cholesterol efflux |
| | GO:0150078 | positive regulation of the neuroinflammatory response |
| | GO:1903980 | positive regulation of microglial cell activation |
| MMU-MIR-181A-5P | GO:0032691 | negative regulation of interleukin-1 beta production |
| | GO:0032715 | negative regulation of interleukin-6 production |
| | GO:0032720 | negative regulation of tumor necrosis factor production |
| | GO:0035195 | gene silencing by miRNA |
| | GO:0050723 | negative regulation of the interleukin-1 alpha biosynthetic process |
| | GO:0050728 | negative regulation of the inflammatory response |
| MMU-MIR-125A-5P | GO:0010629 | negative regulation of gene expression |
| | GO:0035278 | miRNA mediated inhibition of translation |
| | GO:1903671 | negative regulation of sprouting angiogenesis |

Fig 3A illustrates a Venn's diagram which shows that while 365 miRNAs were commonly expressed in both genotypes, 52 miRNAs were exclusively expressed in the WT (Table 3; see also S1 Table), and only four miRNAs belonged exclusively to the TLR4-KO mice (Table 4).

When we looked at the specific miRNAs with higher counts (illustrated in Table 3). We noted that some, such as miR-382-3p, were related to the TLR4/MyD88/NF-κβ signaling pathway [40], and miR-592-5p was involved in brain injury [38]. Notably, these miRNAs were not detected in the TLR4-KO mice by our NGS data analysis. For the specific miRNA expressed in the TLR4-KO mice shown in Table 4, we were unable to find any relevant functions or pathway related to these miRNAs in the bibliography.

To confirm and validate the NGS data, we decided to evaluate the expression of some miRNAs related to inflammatory processes in the WT and TLR4-KO mice cortices by RT-qPCR (the selected miRNAs are depicted in red in Tables 3 and 4). Fig 3B shows the differential expression pattern of the selected miRNAs. Here we observed how miR-382-3p, miR-27b-5p, miR-592-5p and miR-377-3p, which were exclusively found miRNAs in the WT mice, showed a high expression level in the WT mice, but we noted a slight expression for the TLR4-KO mice. These findings demonstrate that RT-qPCR was a more sensitive technique than NGS.

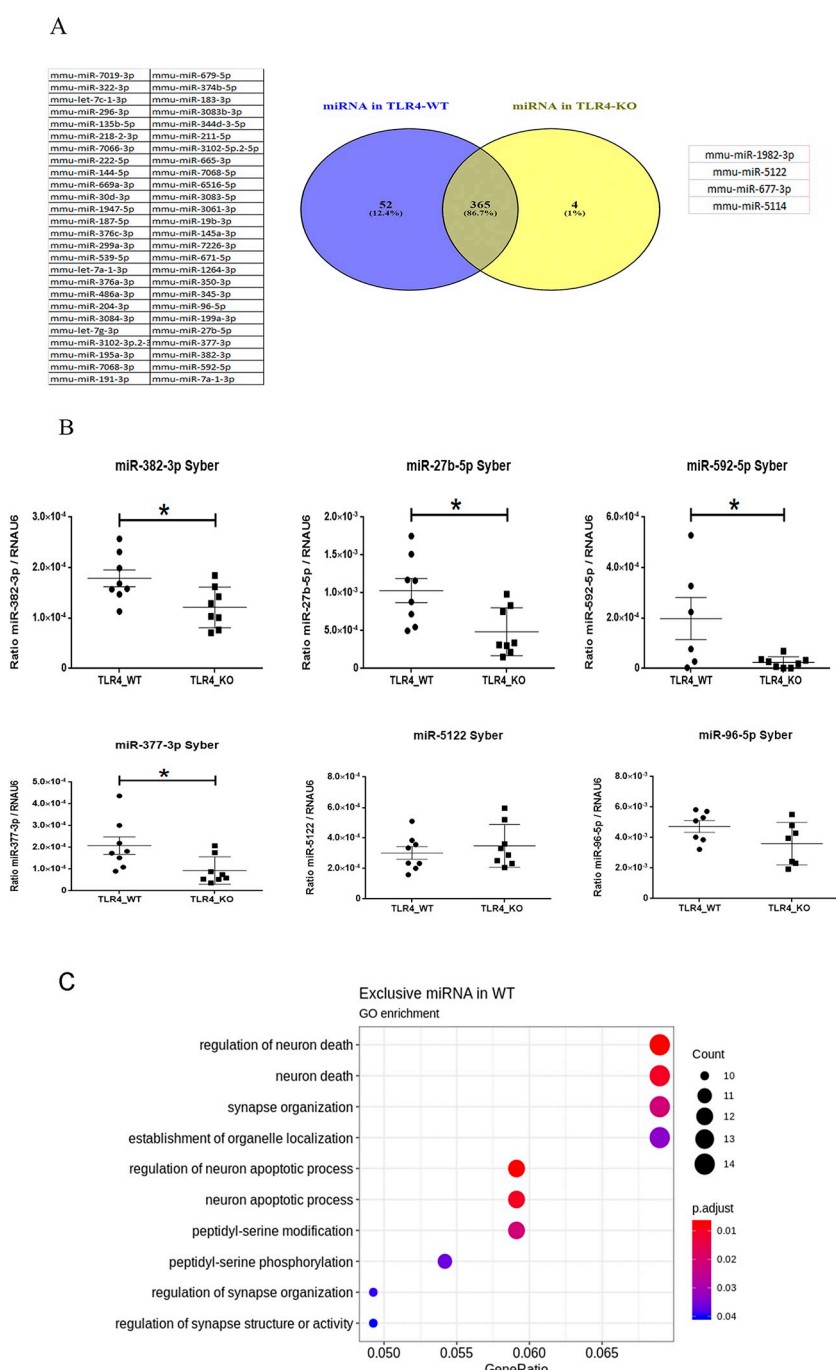

**Fig 3.** Unique miRNA profile for each genotype: (**A**) The Venn's diagram illustrated that 365 miRNAs were commonly expressed in both genotypes; 52 miRNAs were exclusively expressed for the WT and only four miRNAs belonged to the TLR4-KO mice. (**B**) RT-qPCR confirmed the differential expression pattern of some miRNAs, e.g. miR-382-3p, miR-27b-5p, miR-592-5p and miR-377-3p, which were down-regulated in the TLR4-KO mice, while miR-96-5p and miR-5122 were not deregulated as predicted by NGS. A statistical analysis was performed by the t-test with n = 6–8. Statistical significance with a p-value of $^{*}<0.05$ and $^{**}<0.01$. (**C**) Bioinformatics enrichment analysis of specific miRNAs in the WT mice.

**Table 3. Exclusive miRNAs in the TLR4-WT mice.** The table shows the total counts and a selected article about their function.

| MIRNA ONLY IN TLR4-WT | TLR4-WT1 | TLR4-WT2 | TLR4-WT3 | TLR4-KO2 | TLR4-KO3 | LITERATURE | PMID | MATURE SEQUENCE |
|---|---|---|---|---|---|---|---|---|
| MMU-MIR-7A-1-3P | 5 | 10 | 14 | 0 | 0 | Differential stress induced c-Fos expression in the dorsal raphe and amygdala of high-responder/low-responder rats. | [37] | caacaaaucacagucugccaua |
| MMU-MIR-592-5P | 7 | 10 | 9 | 0 | 0 | Effects of MicroRNA-592-5p on hippocampal heuron injury following hypoxic-ischemic brain damage. | [38] | auugugucaauaugcgaugaugu |
| MMU-MIR-377-3P | 7 | 9 | 3 | 0 | 0 | miR-377-3p drives malignancy characteristics by up-regulating GSK-3β expression and activating the NF-κB pathway in hCRC cells. | [39] | aucacacaaaggcaacuuuugu |
| MMU-MIR-382-3P | 6 | 2 | 11 | 0 | 0 | miR-382-3p suppressed the IL-1β induced inflammatory response of chondrocytes byia the TLR4/MyD88/NF-κB signaling pathway. | [40] | ucauucacggacaacacuuuuu |
| MMU-MIR-27B-5P | 3 | 10 | 5 | 0 | 0 | Exosomal microRNA profiles from serum and cerebrospinal fluid in neurosyphilis. | [41] | agagcuuagcugauuggugaac |
| MMU-MIR-199A-3P | 5 | 6 | 3 | 0 | 0 | miR-199a-3p is involved in estrogen-mediated autophagy through the IGF-1/mTOR pathway in osteocyte-like MLO-Y4 cells. | [42] | acaguagucugcacauugguua |
| MMU-MIR-96-5P | 3 | 4 | 7 | 0 | 0 | miR-96-5p prevents hepatic stellate cell activation by inhibiting autophagy via ATG7. | [43] | uuuggcacuagcacauuuuugcu |
| MMU-MIR-345-3P | 2 | 7 | 4 | 0 | 0 | Mitochondria- associated microRNA expression profiling of heart failure. | [44] | ccugaacuaggggucuggagac |

We noted that miR-96-5p, miR-5122 and miR-1982 showed a very low expression in both genotypes, and no significant difference was found when comparing the WT *vs*. The TLR4-KO mice. It is noteworthy that when PCR cycles were used, miR-96-5p and miR-5122 were detected, but miR-1982 was not detected by real-time PCR.

After corroborating that the TLR4-KO genotype presented an altered miRNAs pattern in both quantity and diversity by RT-qPCR, we decided to perform a functional analysis with the 52 miRNAs found exclusively in the WT mice with the mdgsa package (Fig 3A). RT-qPCR provided us with information which indicated that the miRNAs not detected in TLR4-KO by the NGS data were detected by qPCR, but at a very low level compared to the WT mice. The bioinformatics results are offered in Fig 3C. Remarkably this figure depicts a strong regulation on routes like neuron death, apoptosis processes and synapsis organization, among others. These results suggest a lower regulation of TLR4-KO during these important routes/processes than the WT mice.

**Table 4. Exclusive miRNAs in the TLR4-KO mice.** The table shows the total counts and a selected article about their function.

| MIRNA ONLY IN TLR4-KO | TLR4-WT1 | TLR4-WT2 | TLR4-WT3 | TLR4-KO2 | TLR4-KO3 | LITERATURE | PMID | MATURE SEQUENCE |
|---|---|---|---|---|---|---|---|---|
| MMU-MIR-1982-3P | 0 | 0 | 0 | 2 | 2 | no items found | | ucucacccuauguucucccacag |
| MMU-MIR-5122 | 0 | 0 | 0 | 2 | 2 | no items found | | ccgcgggacccgggggcugug |
| MMU-MIR-5114 | 0 | 0 | 0 | 2 | 3 | no items found | | acuggagacggaagcugcaaga |
| MMU-MIR-677-3P | 0 | 0 | 0 | 3 | 2 | no items found | | gaagccagaugccguuccugagaagg |

The analysis of the differential expression of miRNAs in the WT vs. TLR4-KO mice shows a deregulation of the miR-200 family and the miR-99b/let-7e/miR-125a cluster in TLR4-KO

We went on to evaluate the differential expression pattern of the miRNAs in the TLR4-WT vs. TLR4-KO mice. The dendrogram diagram illustrated in Fig 4A shows a marked difference in the expression levels of several miRNA in the WT and TLR4-KO mice cortices. Table 5 also provides complete information on the fold changes, statistics and p-values for the different miRNAs.

When we considered the data in Fig 4A and Table 5, we saw the systematic down-regulation of the family miR-200 components in the TLR4-KO *vs*. WT mice. Fig 4B illustrates the two subfamilies: the first is composed of miR-141 and miR-200c located in mouse chromosome 6 (chromosome 12 in the human genome); the second subfamily includes miR-200a, miR-200b and miR-429, located in chromosome 4 in the mouse genome (chromosome 1 in the human genome).

In the first subfamily, miR-200a-3p and miR-141-3p were down-regulated 5.02- and 4.63-fold in TLR4-KO compared to WT. Other components like miR-429-3p, miR-200a and miR-200b showed a down-regulation of around 4.5-, 5- and 3.5–fold, respectively (Table 5 and Fig 4A). Notably, these miRNAs are associated with the response to LPS activity [45].

With the heatmap (Fig 4A), we also detected two miRNAs that formed part of the cluster formed by miR-99b/let-7e/miR-125a, which showed a higher expression in the TLR4-KO than in the WT mice. Interestingly, this cluster acts as a negative regulation on the TLR signaling pathway by controlling a set of target genes involved in the receptor pathway [46]. However, before assessing the functional analysis of the pathways modulated by these miRNAs, we validated them by RT-qPCR.

Fig 4C shows that all the miRNAs down-regulated by NGS were also down-regulated by RT-qPCR, which indicates that both techniques strongly correlate. The results showed that whereas the miR-200 family components were down-regulated in TLR4-KO, the miRNAs of cluster miR-99b/let-7e/miR-125a, such as miR-99b and mi-125a, tended to be up-regulated. These results indicate that specific miRNAs clusters are deregulated in the TLR4-KO mice cortices compared to the WT samples, which implies a differential miRNA profile related to the genotype.

## Functional analysis of miRNAs shows the major depletion of the inflammatory pathways in the TLR4-KO mice

We next assessed the signaling pathways affected by the deregulation profile of the miRNAs in the cerebral cortices of those mice lacking the TLR4 receptor vs. the WT. For this purpose, a gene set analysis (GSA) was performed along with the Reactome, Biological Processes GO and KEGG pathways. These approaches allowed us to identify etiologic pathways and functional annotations and could, thus, provide novel biological insights [47]. Fig 5 illustrates the GSA results using the bioinformatics and experimental validated targets. The size of the round node was the number of genes in the function. Color was the adjusted p-value.

The Reactome pathways analysis (Fig 5A), run with an inferred bioinformatics targets list, revealed that TLR4-KO was depleted of numerous pathways linked with the immunological system and TLRs signaling. For instance, *cytokines and interleukin signaling and immune system*, "*MyD88 dependent cascade initiated on endosome*", "*Toll-like Receptor 7/8 (TLR 7/8) Cascade*", "*TRAF-6 mediated induction of NF-kβ, MAP kinases upon TLR 7/8 or 9 activation*", etc.

Conversely in the WT group, the analysis showed depleted pathways in relation to the metabolism of inflammatory regulation molecules. The analysis run with the experimental inferred target lists also revealed depleted pathways related to the "Neuronal System" and

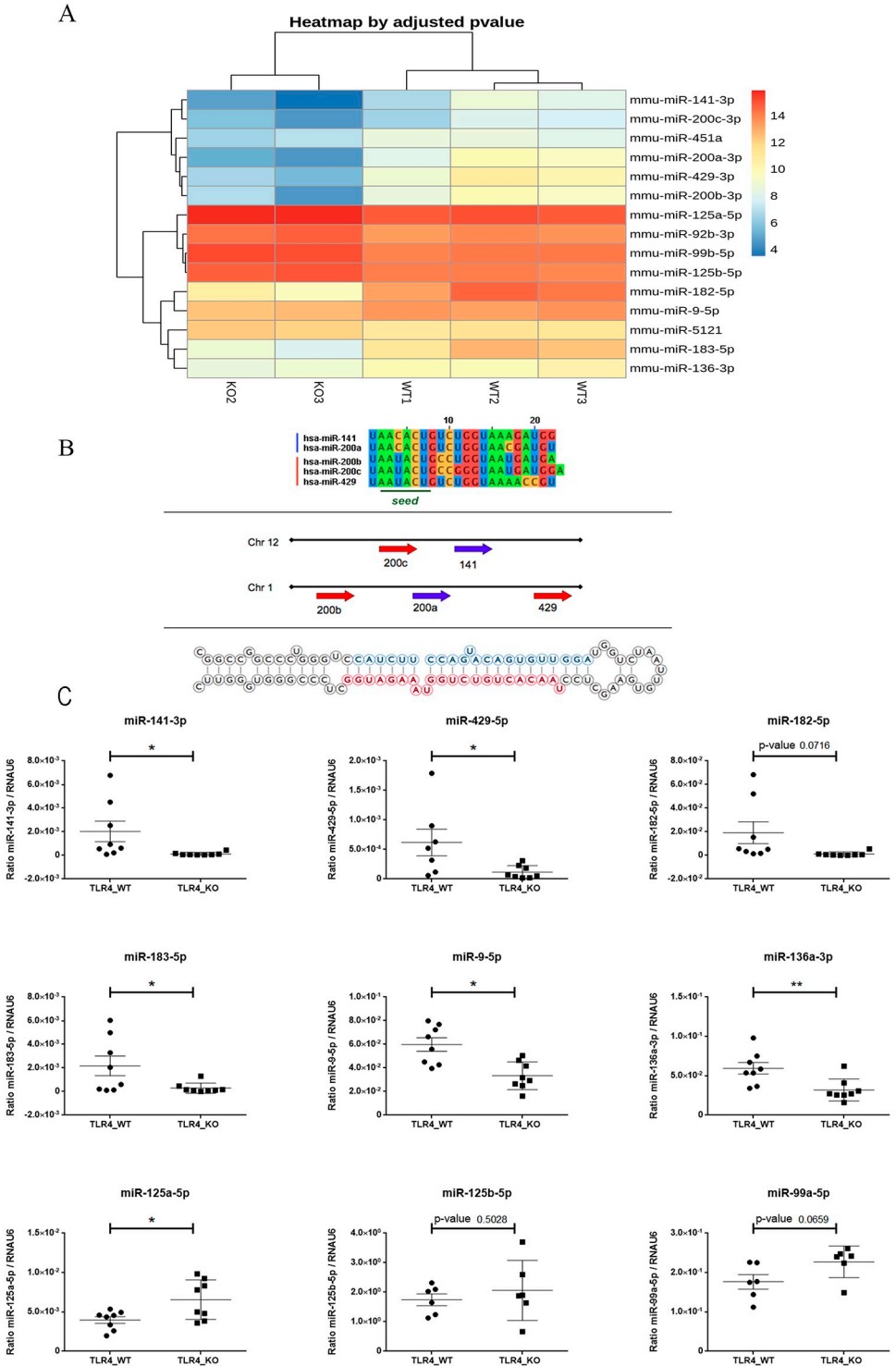

**Fig 4.** Differential expression pattern between the TLR4 and TLR4-KO mice: (**A**) Heatmap diagram of the comparative TLR4 *vs.* TLR4-KO mice. (**B**) Sequence of the five miR-200 family members with the human genomic localization of two clusters containing miR-200 family members; http://atlasgeneticsoncology.org/Genes/GC_MIR141. html. (**C**) RT-qPCR shows the expression of some mature miRNAs selected in WT and TLR4-KO (miR-141-3p, miR429-5p, miR-182-5p, miR-136-3p, miR-9-5p, miR-125a-3p, miR-125b-5p, miR99a-5p) (n = 6–8). The statistical analysis was performed by the Test-t-. Statistical significance is denoted by a p-value of <0.05.

**Table 5. Differential expression results for the miRNAs profiles in the comparative TLR4 *vs*. TLR4-KO mice.**

| miRNA | logFC | logCPM | F | PValue | FDR |
|---|---|---|---|---|---|
| mmu-miR-182-5p | -3.9719505 | 13.512520 | 404.36352 | 0e+00 | 0.00e+00 |
| mmu-miR-183-5p | -3.8092076 | 11.610893 | 244.90941 | 0e+00 | 0.00e+00 |
| mmu-miR-429-3p | -4.2064135 | 9.640591 | 143.75382 | 0e+00 | 0.00e+00 |
| mmu-miR-200a-3p | -5.0255762 | 8.843745 | 116.80459 | 0e+00 | 0.00e+00 |
| mmu-miR-200b-3p | -3.4471567 | 8.943224 | 87.78602 | 0e+00 | 0.00e+00 |
| mmu-miR-141-3p | -4.6305701 | 7.620256 | 58.62150 | 0e+00 | 0.00e+00 |
| mmu-miR-9-5p | -0.9452873 | 13.167504 | 30.57290 | 0e+00 | 6.00e-06 |
| mmu-miR-451a | -1.9476349 | 8.113536 | 28.13968 | 1e-07 | 1.92e-05 |
| mmu-miR-200c-3p | -2.5575429 | 7.183744 | 25.60193 | 4e-07 | 6.54e-05 |
| mmu-miR-136-3p | -1.2688583 | 9.777238 | 25.45641 | 5e-07 | 6.54e-05 |
| mmu-miR-99b-5p | 1.0160077 | 14.624300 | 50.12605 | 0e+00 | 0.00e+00 |
| mmu-miR-125a-5p | 0.9073845 | 15.386571 | 45.41301 | 0e+00 | 0.00e+00 |
| mmu-miR-92b-3p | 0.9775661 | 14.065904 | 42.16099 | 0e+00 | 0.00e+00 |
| mmu-miR-125b-5p | 0.8289091 | 14.400916 | 32.08300 | 0e+00 | 3.00e-06 |
| mmu-miR-5121 | 0.9294424 | 11.644308 | 24.86325 | 6e-07 | 8.30e-05 |

"Potassium channels" in the knockout group, while the pathways associated with carbohydrate metabolism and olfactory signaling were depleted in the WT mice.

Functional profiling for Biological Processes indicated a significant reduction in several processes linked with the neurodevelopment and neuronal synapse processes in the TLR4-KO group, while the responses to the pheromones processes were depleted in the WT group (Fig 5B).

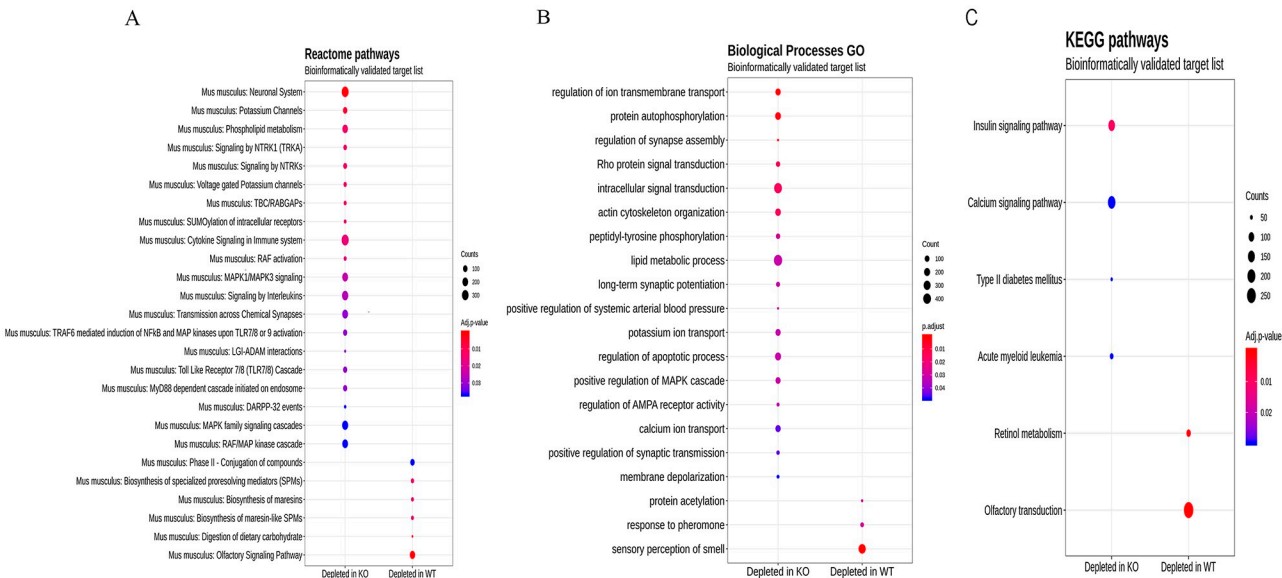

**Fig 5. Enrichment analyses: Enrichment analyses results using miRNA to target the bioinformatics and experimental lists from the miR Database and TargetScan. Analyses were performed with mdgsa [31] and clusterProfiler [32].** (**A**) Reactome pathways. (**B**) Biological Processes GO. (**C**) KEGG pathways. Abscissa axis indicates if the pathway or function is depleted in the WT or the knockout group. Ordinate axis displays the GO term or the pathway. The size of the round node is the number of genes in the function. Color is the adjusted p-value. Raw p-values were adjusted by FDR (Adjusted p-value < 0.05).

Finally, the KEGG pathways analysis demonstrated the activity depletion of diverse diseases routes, like diabetes and acute myeloid leukemia, as well as insulin and calcium signaling pathways, in the knockout group. However, the WT group revealed depleted pathways related to retinol and olfactory transduction routes (Fig 5C).

In short, by the NGS technology and bioinformatics approaches, the results revealed new insights into the molecular and biological processes associated with protective effects of TLR-KO against neuroinflammation, brain damage neurodegeneration and alcohol abuse.

## Functional enrichment analysis of Toll-like Receptors 7/8

To further elucidate the hierarchical functions of miRNAs in gene regulatory networks in TLR4-KO vs. WT mice, we next performed a complementary and enrichment analysis of some routes obtained by the bioinformatics analysis. We selected *Toll-like Receptor 7/8 (TLR 7/8)* because these receptors have been related to TLR4 pathways and alcohol abuse [48]. We therefore performed a RT-qPCR analysis of TLR7 and TLR8, and of some genes associated with their signaling response, such as IL12 and INFα. Fig 6 shows that while ethanol, an inflammatory activator, increased the TLR8 mRNA levels in both genotypes, no changes in

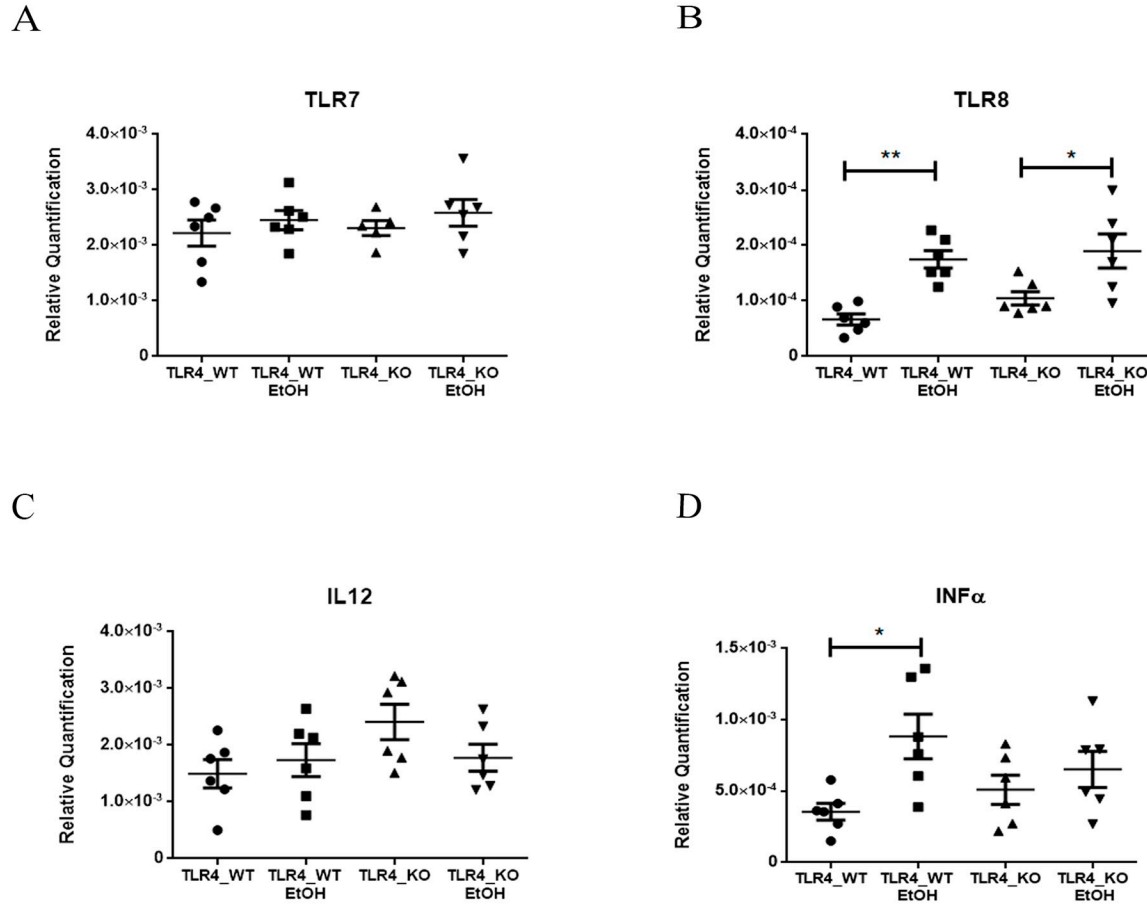

**Fig 6. Lack of TLR4 dismisses INFα activation by TLR7/8 Receptors. (A-D)** The RT-qPCR analysis of the WT and TLR4-KO mice cortices with and without EtOH treatment, the relative quantification of receptors TLR7/8, Interleukin IL12 and INFα showed a differential expression of TLR8 by the ethanol inflammatory effect in both genotypes, but alcohol led to the activation of INFα only in the WT mice. A statistical analysis was performed by a two-way ANOVA with n = 6. Statistical significance with a p-value of $^{*}$<0.05 and $^{**}$<0.01.

gene expression were noted in TLR7. With the downstream genes, while no changes were noted in Il12, INFα mRNA significantly increased with ethanol in the WT mice, but not in TR4-KO.

## Discussion

TLR4 receptors are highly conserved receptors that play a key role in the innate immune system by recognizing microbial subproducts, such as pathogen-associated molecular pattern and endogenous molecules, which results in inflammatory response activation [49, 50]. The activation of these receptors induces intracellular signaling by the myeloid differentiation factor 88 (MyD88) dependent and MyD88-independent pathways. The final event induces the activation of the downstream transcription factors, such as NF-κβ, among others, involved in cellular responses and the production of proinflammatory cytokines [51].

Recent studies have demonstrated the importance of TLR4 receptors in both metabolic functions and neuropathology because the absence of these receptors prevents metabolic damage related to fatty acids [52], ischemic injury [53] or neuroinflammatory, neurodegenerative and demyelination disorders [54, 55]. Our previous studies have shown the crucial participation of the TLR4 signaling response in ethanol-induced brain damage, neurodegeneration and behavioral dysfunction alteration [56, 57], effects which were not observed in alcohol-treated TLR4-KO mice. By NGS and bioinformatics tools, we recently identified miRNAs that were differentially expressed in chronic alcohol-treated *vs*. untreated WT or TLR4-KO mice cortices. In particular, we demonstrated the differential expression of the miR-183 cluster (miR-96/-182/-183), miR-200a and miR-200b, which were down-regulated, while mirR-125b was up-regulated in the alcohol-treated WT *versus* (*vs*.) untreated mice [17]. The functional enrichment of the miR-183C and miR-200a/b family target genes revealed that neuroinflammatory pathways networks are involved in TLR4 signaling associated with alcohol abuse. Notably, the neuroinflammatory target genes were abolished in TLR4-KO mice. These results raise the question about which miRNAs, genes or signaling proteins are down-regulated in TLR-KO to be able to exert their protective effects.

Here our results demonstrate that the TLR4-KO mice presented a significant lack or down-regulation in several mature miRNAs involved in the inflammatory response compared to the profile observed in the WT mice. Indeed the Venn's diagram (Fig 2A) shows that 52 miRNAs were exclusively expressed in the WT mice, but these miRs had a very low expression in TLR4-KO, as demonstrated by RT-qPCR. Some of these down-regulated miRNAs, such as miR-382-3p, are involved in the regulation of IL-1β and TLR4 signaling inhibition [40], while miR-27b-5p participates in microbial infection defense [58]. The Venn's diagram also illustrates that four miRNAs were exclusively expressed in TLR4-KO (miR-1982-3p, miR-5122, miR-677-3p, miR-5114). However, the information, functions and validated targets of this set of miRNAs are practically inexistent and only miR-5122 was detected by RT-qPCR with a very low expression. These results indicate that lack of TLR4 significantly alters the number and expression of several miRNAs. Indeed our previous studies demonstrated that ethanol treatment dysregulated miR-200s family and target genes (e.g. MAPK, IL1R1) in the WT, but not in the TLR4-KO mice treated with alcohol (17). Moreover, we showed that the miR-200 family was significant down-regulated in the TLR4-KO mice, which supports the absence of neuroinflammatory effects of ethanol on mice lacking this receptor.

The functional analysis of the miRNAs exclusively expressed in the WT mice cortices (Fig 3C) provided information on the role of these miRs in neuron death, apoptosis processes, synapsis organization, peptidyl serine signaling or synapsis structure and activity. Interestingly, the down-regulation of these miRNAs in the TLR4-KO mice offered some protection from

neuronal death and brain injury in different pathologies. For instance, lack of TLR4 can protect mice from ischemic brain injury [59], brain damage/inflammation after experimental stroke [60], neuroinflammation, brain injury [15], Parkinson's disease [61] and demyelination associated with alcohol abuse [16]. It is important to stress that the elimination of TLR4 not only abolished the activation of glial cells and the production of inflammatory cytokines in the brain, but could also block the recruitment of peripheral inflammatory cells and increase BBB permeability by participating in some neurodegenerative diseases [62].

From the Venn's diagram, we also detected 362 miRNAs that were commonly expressed in both genotypes. Using the heatmap to assess the differential expression of the miRs in TLR4 vs. TLR4-KO (Fig 4A), we revealed the presence of different miRNAs belonging to the miR-200 family. All the components of this miRNAs family showed a down-regulation (miR-141 and miR-200a and miR-200b, miR-200c, miR-429) in TLR4-KO vs. WT. Interestingly, miR-200 components like miR-200b and miR-200c, alter the efficiency of TLR4 signaling through the MyD88-dependent pathway, and lack of the TLR4 receptor can modify miR-200 family levels by drastically lowering their expression [63]. Similarly, miR-141 and miR-200c regulate inflammation, while loss of miR-141/200c significantly reduces inflammation, hepatic steatosis and injury in non-alcoholic fatty livers in mice [64]. MiR-141-3p is also involved in chronic inflammatory pain [65], and miR-141 inhibition improves mortality, neurological deficits, and decreased infarct volumes during post-stroke social isolation in aged mice [66]. Likewise, miR-141 and other miR-200 family members also regulate oxidative stress by targeting the p38 (MAPK14) transcript [67, 68]. The heatmap data also demonstrated the up-regulation of the two components of the miR-99b/let-7e/miR-125a cluster (miR99b and miR-125a) in TLR4-KO vs. WT, whereas some studies have demonstrated the anti-inflammatory role of this cluster, which influences the production of pro-inflammatory cytokines in response to LPS [46].

The Functional Profiling for Biological Processes identified the pathways that were affected by the specific miRNA profile in each genotype. In general, inflammatory pathways are attenuated in the TLR-KO genotype [69], such as cytokine and interleukin signaling and inflammatory processes, as demonstrated after brain injury or diabetes in TLR4-KO [70, 71].

One interesting result was that lacks of TLR4 induced the depletion of other TLRs, like TLR7/8. These receptors are located in endosomal membranes and recognize single pathogenic stranded RNAs (ssRNA) and DNAs. TLR7/8 receptors have been associated with innate immunity to autoimmunity [72, 73]. Some studies also suggest that ethanol exposure increases TLR7 expression and the release of let-7b in microglia-derived microvesicles, which contributes to neuroimmune gene induction [74]. The functional analysis of these receptors demonstrated that while ethanol increased the TLR8 mRNA levels in both genotypes, no changes in gene expression were noted in TLR7 when comparing WT and TLR4-KO.

Other studies have demonstrated the role of miR-1906 as an important target of the TLR4 receptor in ischemic damage [75]. We detected miR-1906 in our samples, but with a very low expression. Other authors have suggested the role of either miR-182-5p in the neuroprotective effect of cerebral ischemia-reperfusion injury via regulation by TLR4 [76] or miR-124 and miR-146a in reversing neuropathic pain through the TLR4 receptor [77].

In short, the present study provides insights into the role of miRs associated with the TLR4 response. This receptor modulates and activates the inflammatory signaling response to trigger inflammation and tissue damage as lack of these receptors (TLR4-KO) significantly blocks or reduces the expression of the miRs associated with the inflammatory signaling response by contributing to immune disorders, including neurodegeneration and ischemic damage.

## Supporting information

**S1 Table.**
(PDF)

**S1 Raw images.**
(TIF)

## Acknowledgments

We are grateful to Dr. S. Akira, who provided us with the TLR4−/− mice, and to M. J. Morillo Bargues and R. Lopez Hidalgo for their excellent technical assistance.

## Author Contributions

**Conceptualization:** Juan R. Ureña-Peralta, Francisco García-García, Consuelo Guerri.

**Data curation:** Raúl Pérez-Moraga.

**Formal analysis:** Juan R. Ureña-Peralta, Francisco García-García.

**Funding acquisition:** Consuelo Guerri.

**Methodology:** Juan R. Ureña-Peralta, Raúl Pérez-Moraga.

**Supervision:** Juan R. Ureña-Peralta, Francisco García-García, Consuelo Guerri.

**Validation:** Juan R. Ureña-Peralta, Francisco García-García, Consuelo Guerri.

**Writing – original draft:** Juan R. Ureña-Peralta, Raúl Pérez-Moraga, Francisco García-García, Consuelo Guerri.

**Writing – review & editing:** Consuelo Guerri.

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
