## [Decision Letter · Decision Letter 0]

21 Apr 2020

PONE-D-19-31787

Lack of TLR4 modifies the miRNAs profile and attenuates inflammatory signaling pathways

PLOS ONE

Dear DR Guerri,

Thank you for submitting your manuscript to PLOS ONE. After careful consideration, we feel that it has merit but does not fully meet PLOS ONE’s publication criteria as it currently stands. Therefore, we invite you to submit a revised version of the manuscript that addresses all of the points raised during the review process.

We would appreciate receiving your revised manuscript by Jun 05 2020 11:59PM. To enhance the reproducibility of your results, we recommend that if applicable you deposit your laboratory protocols in protocols.io, where a protocol can be assigned its own identifier (DOI) such that it can be cited independently in the future. For instructions see: http://journals.plos.org/plosone/s/submission-guidelines#loc-laboratory-protocols

We look forward to receiving your revised manuscript.

Kind regards,

Pratibha V. Nerurkar, Ph.D

Academic Editor

PLOS ONE

Journal Requirements:

Reviewers' comments:

Reviewer's Responses to Questions

**Comments to the Author**

1. Is the manuscript technically sound, and do the data support the conclusions?

Reviewer #1: Yes

Reviewer #2: Yes

Reviewer #3: Yes

2. Has the statistical analysis been performed appropriately and rigorously? 

Reviewer #1: I Don't Know

Reviewer #2: Yes

Reviewer #3: Yes

3. Have the authors made all data underlying the findings in their manuscript fully available?

Reviewer #1: Yes

Reviewer #2: Yes

Reviewer #3: Yes

4. Is the manuscript presented in an intelligible fashion and written in standard English?

Reviewer #1: No

Reviewer #2: Yes

Reviewer #3: Yes

5. Review Comments to the Author

Reviewer #1: In the article entitled “Lack of TLR4 modifies the miRNAs profile and attenuates inflammatory signaling pathways,” by Urena-Peralta et al., the authors used the cerebral cortex from TLR4 KO and wildtype mice with next generation sequencing to determine how TLR4 regulates miRNA.

Major Concerns:

1) Why were female mice only used? Any role of the estrogen cycle?

2) Discussion is a bit long

Minor Concerns:

1) Please have a native English speaker edit for language

Reviewer #2: Comments to author

In the present study Ureña-Peralta and colleagues investigated the role of Toll-like 4 receptors (TLR4s) for the expression of microRNAs relevant to inflammatory pathways in mouse cerebral cortex. The authors utilized an experimental strategy, where cortices of transgenic TLR4 KO mice and WT controls, were analyzed with next generation sequencing targeting specific microRNAs, followed by validation of microRNA expression using qPCR. The authors first demonstrate the effectiveness of the global TLR4 KO approach on attenuating long-term EtOH-induced inflammatory markers (e.g. IL-1R, IL-1beta, IL-6 and COX2). They subsequently, found that absence of TLR4 significantly downregulated miR-382-3p, miR-27b-5p, miR-592-5p and miR-377-3p, as predicted by NGS; Some of these targets are relevant for TLR4/MyD88/NF-κB signaling and brain injury. Further, the authors investigated the differential expression pattern of the miRNAs and found that TLR4-KO mice exhibit downregulation of the miR-200 family and upregulation of the miR-99b/let-7e/miR-125a cluster. The authors further employed bioinformatics and functional analysis for exploring the role of the specific miRNA profile of the WTs vs TLR4-KOs on central molecular pathways involved in neuroinflammation, and found depletion of cytokine and interleukin signaling, MAPK and ion Channels routes, MyD88 pathways, NF-κB and TLR7/8 pathways. Though the comparative strategy of this study is elegant and the results are novel, additional experiments are needed to support the authors claims.

The sample-size used for the NGS analysis is very small (n=3 WT vs n=2 TLR4 KOs). This is a big concern as NGS technologies, as all other biotechnologies, are affected by both biological and technical variability. It would be beneficial to increase the n-number of the groups to gain statistical power and achieve comparability with your very clear qPCR data.

In your methods section, please provide details on brain extraction and dissection: Did you perfuse the brains? Did you use a brain matrix? What were your estimated coordinates and the thickness of the brain section?

It would be relevant to include in your discussion a comparison to the findings of your previous TLR4 study (Ureña-Peralta et al., 2018), as some of the miRNAs uncovered are similar.

Please discuss the limitations of the global TLR4 KO strategy: Are you expecting these effects to be due to CNS TLR4s or secondary effects due to peripheral immune-to-brain signaling, or blood-brain-barrier endothelial TLR4s?

General manuscript comments:

The image resolution of the figures is very poor! It in many cases not possible to read the figure labels. It makes it extremely difficult to evaluate consistency between what is written in the results-section and the actual data - Please revise.

Several of the figure-legends are incomplete, for instance some are missing n-numbers, while the figure legend for figure 1 is unstructured and missing (C) and (D). Please revise the figure-legends.

The language quality varies throughout the individual sections of the manuscript. The NGS-methods and discussion-sections are very well written, while the introduction has lacks (see minor comments below). Please make the language more coherent.

Language, typos and understanding corrections:

Please correct PAMPs from pattern associated molecular pattern to pathogen associated molecular pattern page 3 line 3.

Please revise the sentence: “The sequence-complementary mechanism of miRNA activity exploits combinatorial regulate the expression of genes by repressing the translation of their complementary target genes (9)”. Page 3, section 3, line 2. Do you mean: “….activity exploits combinatorial regulation of gene expression by….”?

Please revise aim 2 on page 4, it is unclear what you mean: “2) if there are compensatory changes in certain miRNAs that could explain the lack of effects with the neuroinflammation associated with alcohol intake”.

In the introduction you refer to your previous study (ref 17), which clearly is linked to this aim, however you don’t specify what the “lack of effects” were.

Furthermore, based on this study a question was raised “These results raise the question as to what the impact of miRNAs profile in mice is lacking TLR4 response?”. If this question is linked to aim 2, it needs to be further clarified, also it is unclear what you mean – please revise the question.

On page 5, under the Alcohol treatment section, line 10-11: You have written “obtained” several times, hence the sentence has become fragmented, also please state the actual BAL-values as this is useful information for the reader.

On page 10 under the statistical section, line 2, Western blotting results are mentioned. However, there is no description of such method or data elsewhere in the manuscript.

On page 19, line 1: “Figure 4B, obtained from, illustrate the two subfamilies…”, please specify what it was “obtained from” or simply delete the obtained from, it is unclear what you mean.

Reviewer #3: Summary

TLR4 is a member of toll-like receptors (TLRs). Lack of TLR4 has been shown to protect against inflammatory processes, neuroinflammation cause by chronic alcohol consumption. The authors aimed to understand whether the reduced inflammation in the cortex of TLR4 knock out mice was mediated by an alteration in the expression of miRNAs involved in inflammatory pathways. Using a next-generation sequencing (NGS) approach they screened the cerebral cortex of TLR4 knock out mice and wild type controls to identify miRNAs that were differentially regulated in TLR4 knock out mice and controls. Understanding how TLR4 mediates neuroinflammation in chronic alcohol consumption is very important because it could shed light on ways to modulate these processes and prevent brain damage.

Critique

1. The manuscript is technically sound and the statistical analysis has been performed appropriately and rigorously. The results obtained by sequencing are always validated by RT-PCR and the appropriate controls are included in each figure. Moreover the data support the conclusions.

2. One major concern is the observation that TLR4KO mice show depletion of TLR7/8. How can you be sure that the micro-RNA differential expression is not due to the lack of TLR7/8. Do TLR7/8 mice show micro-RNA-dysregulation?

3. The authors could focus on a few relevant micro-RNAs involved in inflammation that are differentially expressed and identify the effectors of the TLR4 signaling pathway that modulate these micro-RNAs.

4. The authors could then exogenously modulate the expression (overexpression or knockdown) or function of specific effectors in TLR4 KO mice to rescue the expression of specific micro-RNAs

5. While the statistical analysis has been performed appropriately, the description of the statistical analysis should be described for each panel (it is for example missing in Figure 3b).

6. Figure resolution is low and figure layout can be improved. In Figure 4, for example, some panels are extremely big (Figure 4a) while others are barely visible (Figure 4b).

7. Figure 1: Figure 1C-D description is missing in the figure legend.

8. Figure 4b: Specify that the chromosomal localization refers to the human genome both in the legend and the main text, where there is only reference to the mouse genome, which could create confusion.

6. PLOS authors have the option to publish the peer review history of their article (what does this mean?). If published, this will include your full peer review and any attached files.

Reviewer #1: No

Reviewer #2: No

Reviewer #3: No

---

## [Author Response · Author response to Decision Letter 0]

25 Jun 2020

Answer to reviewers

Reviewer #1:

Major Concerns:

1) Why were female mice only used? Any role of the estrogen cycle?

Thank you very much for you comment. We used female mice as previous results from our laboratory demonstrated that females are more sensitive to the neuroinflammatory effects of alcohol effects than males (Alfonso-Loeches et al., 2013; Pascual et al., 2017). A comment has been included on p. 5. Another reason was the price to perform next-generation sequencing (NGS). In fact we initially programmed to perform NGS in the cortices of males and females, but the prices to do NGS were twice as high as we had to add four additional groups (WT, WT + Ethanol, TLR4-KO, TLR4-KO +ethanol).

Regarding the effects of the estrogen cycle, although they are very important and could mediate some effects, we did not consider them in the present study.

2) Discussion is a bit long.

As suggested, we have attempt to reduce the Discussion but, at the same time, we have included some comments in response to the reviewers, which made the Discussion longer.

Minor Concerns:

1) Please have a native English speaker edit for language.

Thank you very much for your comment. The manuscript has been revised by a native English speaker.

Reviewer #2: Comments to author.

1) The sample-size used for the NGS analysis is very small (n=3 WT vs n=2 TLR4 KOs). This is a big concern as NGS technologies, as all other biotechnologies, are affected by both biological and technical variability. It would be beneficial to increase the n-number of the groups to gain statistical power and achieve comparability with your very clear qPCR data.

Thank you very much for your comment. As stated in the Material and Methods section (p. 7), each sample is a pool from three different mice cortices and, therefore, the three pools in the WT is the average of nine different cortices. However in TLR4 KO, we had problems with one pooled sample and we used only two pooled samples that were the average of six mice cortices. We have modified the section RNA Quantity and quality determinations paragraph to make it clearer (p. 7-8).

2) In your methods section, please provide details on brain extraction and dissection: Did you perfuse the brains? Did you use a brain matrix? What were your estimated coordinates and the thickness of the brain section?

Thank you very much for your question. On p. 5 we have described details of the procedure used for brain extraction and cortices dissection, and how cortices were processed. Indeed we have added a new section Cerebral cortex dissection (p. 5) in which we describe the specific procedure used. Cortices dissection was performed by the same person using atlas coordinates. Cortices were weighed and immediately snap-frozen in liquid nitrogen until used for further determination analyses. The weight of cortices was similar in all the groups.

3) It would be relevant to include in your discussion a comparison to the findings of your previous TLR4 study (Ureña-Peralta et al., 2018), as some of the miRNAs uncovered are similar.

As suggested, we have discussed and compared previous findings to the present results. In particular, we have discussed the relevance of some miRNAs that were dysregulated in the ethanol-treated WT mice, but not in TLR4-KO mice, as lack of TLR4 receptors down-regulates these miRNAs pathways (p. 24). 

4) Please discuss the limitations of the global TLR4 KO strategy: Are you expecting these effects to be due to CNS TLR4s or secondary effects due to peripheral immune-to-brain signalling, or blood-brain-barrier endothelial TLR4s?

As suggested, we have discussed both the limitation of the TLR4 KO strategy to study neuroinflammation, and the potential involvement of peripheral immune-to-brain signaling in neuroinflammation (p. 24)

General manuscript comments:

5) The image resolution of the figures is very poor! It in many cases not possible to read the figure labels. It makes it extremely difficult to evaluate consistency between what is written in the results-section and the actual data. Please revise.

We appreciate your comment and have included the images in TIFF. We hope that they it is now easier to evaluate and observe the details and figure labels.

6) Several of the figure-legends are incomplete, for instance some are missing n-numbers, while the figure legend for figure 1 is unstructured and missing (C) and (D). Please revise the figure-legends.

Thank you very much for drawing our attention, and we apologize for these mistakes. We have revised all the figures legends and included the correct numbers. Figure 1 has been arranged.

7) The language quality varies throughout the individual sections of the manuscript. The NGS-methods and discussion-sections are very well written, while the introduction has lacks (see minor comments below). Please make the language more coherent.

We thank the reviewer for drawing our attention and have corrected the unclear text and sentences.

-Language, typos and understanding corrections:

-Please correct PAMPs from pattern associated molecular pattern to pathogen associated molecular pattern page 3 line 3.

Thank you very much. We have corrected all the typos.

-Please revise the sentence: “The sequence-complementary mechanism of miRNA activity exploits combinatorial regulate the expression of genes by repressing the translation of their complementary target genes (9). Page 3, section 3, line 2. Do you mean: (“…activity exploits combinatorial regulation of gene expression by…”?).

These sentences have been revised and appropriately changed.

-Please revise aim 2 on page 4, it is unclear what you mean: “2) if there are compensatory changes in certain miRNAs that could explain the lack of effects with the neuroinflammation associated with alcohol intake”.

Thank you, they have been corrected.

-In the introduction you refer to your previous study (ref 17), which clearly is linked to this aim, however you don’t specify what the “lack of effects” were.

We have clarified “lack of effects”.

-Furthermore, based on this study a question was raised “These results raise the question as to what the impact of miRNAs profile in mice is lacking TLR4 response?”. If this question is linked to aim 2, it needs to be further clarified, also it is unclear what you mean – please revise the question.

The question in aim 2 has been changed to a more specific aim.

-On page 5, under the Alcohol treatment section, line 10-11: You have written “obtained” several times; hence the sentence has become fragmented, also please state the actual BAL-values as this is useful information for the reader.

The sentence has been rewritten. Thank you for drawing our attention to this.

-On page 10 under the statistical section, line 2, Western blotting results are mentioned. However, there is no description of such method or data elsewhere in the manuscript.

On page 10, the sentence of the “Western blotting results” has been eliminated.

-On page 19, line 1: “Figure 4B, obtained from, illustrate the two subfamilies…”, please specify what it was “obtained from” or simply delete the obtained from, it is unclear what you mean.

As suggested, we have deleted “obtained from”.

We would like to thank the reviewer for his/her help and suggestions to clarify some parts of the text. All the comments have been taken into account and corrections made.

Reviewer #3: Summary

Critique

1) The manuscript is technically sound and the statistical analysis has been performed appropriately and rigorously. The results obtained by sequencing are always validated by RT-PCR and the appropriate controls are included in each figure. Moreover, the data support the conclusions.

2) One major concern is the observation that TLR4KO mice show depletion of TLR7/8. How can you be sure that the micro-RNA differential expression is not due to the lack of TLR7/8? Do TLR7/8 mice show micro-RNA-dysregulation?

Thank you very much for your appropriate comment. We have performed an additional experiment to evaluate the gene expression levels of receptors TLR7 and TLR8, as well as two genes involved in this pathway. In Fig. 6 we now show that while ethanol increased the TLR8 mRNA levels in both genotypes, no changes in gene expression were noted in TLR7 compared to WT and TLR4KO (see Fig. 6, the results on p. 22 and the Discussion on p. 25 ).

.3) The authors could focus on a few relevant micro-RNAs involved in inflammation that are differentially expressed and identify the effectors of the TLR4 signalling pathway that modulate these micro-RNAs.

Thank you for your comment; the genes analysed in figure 1 (Il1R1, Traf6 and IL6) in addition to the INFα, analyzed in the new figure (Fig. 6), are related with the miR200 family. We hope you have resolved your doubts.

4) The authors could then exogenously modulate the expression (overexpression or knockdown) or function of specific effectors in TLR4 KO mice to rescue the expression of specific micro-RNAs.

We appreciate your comment. In fact although we have attempted to exogenously modulate the expression of TLR4-KO in cortical glial cells in culture from the TLR4-KO mice, we still obtained no clear conclusions.

5) While the statistical analysis has been performed appropriately, the description of the statistical analysis should be described for each panel (it is for example missing in Figure 3b).

We appreciate your comment and have included the statistical analysis in all the figure legends.

6). Figure resolution is low and figure layout can be improved. In Figure 4, for example, some panels are extremely big (Figure 4a) while others are barely visible (Figure 4b).

Thank you very much for your comment. We have provided figures at higher resolutions. The panels for Fig. 4A and 4B have been modified.

7). Figure 1: Figure 1C-D description is missing in the figure legend.

We apologize for this error. We have modified the legend of Figure 1.

8). Figure 4b: Specify that the chromosomal localization refers to the human genome both in the legend and the main text, where there is only reference to the mouse genome, which could create confusion.

Thank you very much for your useful comment. We have now modified the text according to your suggestions.

---

## [Decision Letter · Decision Letter 1]

21 Jul 2020

Lack of TLR4 modifies the miRNAs profile and attenuates inflammatory signaling pathways

PONE-D-19-31787R1

Dear Dr. Guerri,

We’re pleased to inform you that your manuscript has been judged scientifically suitable for publication and will be formally accepted for publication once it meets all outstanding technical requirements.

Kind regards,

Pratibha V. Nerurkar, Ph.D

Academic Editor

PLOS ONE

Additional Editor Comments (optional):

Reviewers' comments:

Reviewer's Responses to Questions

**Comments to the Author**

1. If the authors have adequately addressed your comments raised in a previous round of review and you feel that this manuscript is now acceptable for publication, you may indicate that here to bypass the “Comments to the Author” section, enter your conflict of interest statement in the “Confidential to Editor” section, and submit your "Accept" recommendation.

Reviewer #1: All comments have been addressed

Reviewer #2: All comments have been addressed

Reviewer #3: All comments have been addressed

2. Is the manuscript technically sound, and do the data support the conclusions?

Reviewer #1: Yes

Reviewer #2: (No Response)

Reviewer #3: (No Response)

3. Has the statistical analysis been performed appropriately and rigorously? 

Reviewer #1: Yes

Reviewer #2: (No Response)

Reviewer #3: (No Response)

4. Have the authors made all data underlying the findings in their manuscript fully available?

Reviewer #1: Yes

Reviewer #2: (No Response)

Reviewer #3: (No Response)

5. Is the manuscript presented in an intelligible fashion and written in standard English?

Reviewer #1: Yes

Reviewer #2: (No Response)

Reviewer #3: (No Response)

6. Review Comments to the Author

Reviewer #1: the author should add sentence should be added that estrogen is a potential confounding variable in interpretation.

Reviewer #2: (No Response)

Reviewer #3: (No Response)

7. PLOS authors have the option to publish the peer review history of their article (what does this mean?). If published, this will include your full peer review and any attached files.

Reviewer #1: No

Reviewer #2: No

Reviewer #3: No

---

## [Editor Report · Acceptance letter]

27 Jul 2020

PONE-D-19-31787R1 

Lack of TLR4 modifies the miRNAs profile and attenuates inflammatory signaling pathways 

Dear Dr. Guerri:

I'm pleased to inform you that your manuscript has been deemed suitable for publication in PLOS ONE. Congratulations! Your manuscript is now with our production department. 

Kind regards, 

on behalf of

Dr. Pratibha V. Nerurkar 

Academic Editor

PLOS ONE